# A simple dissection method for the isolation of mouse trabecular meshwork cells

**Maximilian Binter, Fridolin Langer, Xiaonan Hu, Migle Lindziute, Carsten Framme, Jan Tode, Heiko Fuchs** [ORCID] *

Institute of Ophthalmology, Hannover Medical School, University Eye Hospital, Hannover, Germany

* fuchs.heiko@mh-hannover.de

## Abstract

### Purpose

The outflow pathway, especially trabecular meshwork (TM), plays an essential role in glaucoma, and the availability of TM cells is crucial for *in vitro* research. So far, the isolation of TM cells from mice has been anything but manageable due to the small size of the eye. Direct isolation using a stereomicroscope and forceps requires a high grade of dexterity. Indirect isolation is based on the phagocytic properties of TM cells and involves injecting magnetic microspheres into the anterior chamber of live mice followed by isolation. Therefore, a simpler, less expensive, and nonexperimental strategy for isolating mouse TM cells would be desirable.

### Methods

After enucleation, the eyes were cut in half anterior-to-posteriorly. The lens and posterior segment were removed. Iris and the attached ciliary body were gently pulled backward and disconnected from the remaining tissue to expose the TM. By incising through the cornea anteriorly and posteriorly of the TM, the cornea/TM stripe could be isolated. The cornea/TM stripe was cultured with the pigmented side down in a 6-well. The outgrowing pigmented cells were analyzed by immunocytochemistry and mRNA expression for previously described TM cell markers. The phagocytic properties of the cells were additionally confirmed using fluorescent microspheres.

### Results

Pigmented phagocytic cells were the first to grow out of the cornea/TM strips after approximately 4–7 days. Cells were positive for Collagen IV, Fibronectin1, Vimentin, and Actin alpha 2 and could phagocytize fluorescent microbeads. Cross-linked actin networks were visible after 9 days of exposure to TGFB2 (transforming growth factor-beta 2). Additionally, treatment with 500 nM Dexamethasone for one week increased myocilin expression, as previously reported for TM cells. In addition, we proved that this method can also be used in albino mice, which lack pigmentation of the trabecular meshwork.

**Data Availability Statement:** All relevant data are within the manuscript and its Supporting Information files.

**Funding:** The author(s) received no specific funding for this work.

**Competing interests:** The authors have declared that no competing interests exist.

## Conclusions

The isolated cells show phagocytic properties and specific expression of markers reported in TM cells. Therefore, our dissection-based method is inexpensive and reproducible for isolating TM cells in mice.

## Introduction

Glaucoma is the leading cause of irreversible blindness worldwide. Primary open-angle glaucoma (POAG) is the most common type of this disease, with a global prevalence of around 3.1% [1]. With elevated intraocular pressure (IOP) being the major risk factor, the outflow pathway and its resistance play an essential role in this disease. The main factor for outflow resistance is the trabecular meshwork (TM). This sponge-like tissue bridges the Schlemm canal and consists of several cell layers and connective tissue beams. Altered TM cells elevate outflow resistance, leading to increased IOP and POAG. This pathomechanism puts TM cells at the focus of glaucoma research [2]. Frequently used TM cells in studies are human, bovine, and porcine due to their easy accessibility [3]. However, many recent glaucoma mouse models have been developed and are frequently used [4–10]. This popularity is based on the biological similarities of the murine and human eye, the practicability of experiments, the availability of genetic tools in mice, and the relatively low costs [9]. Nevertheless, extracting TM cells from mice is difficult due to the organ size, which results in a complex and resource-intensive process to gain murine TM cells for *in vitro* pilot experiments [11].

Several criteria are used to identify TM cell cultures in any species. TM cells vary in appearance; therefore, morphology alone is an inaccurate criterion. TM cell isolation from mice can be challenging because of potential contamination by neighboring cells, such as Schwalbe's line cells, corneal keratocytes, scleral spurs cells, sclera fibroblasts, or ciliary muscle cells [3]. Therefore, immunocytochemical (ICC) staining is crucial for identifying TM cells. For TM cell identification, markers like collagen IV (COL4), fibronectin 1 (FN1), vimentin (VIM), and actin alpha 2 (ACTA2) have been used [3, 12]. Another more reliable marker is myocilin (MYOC) induction by dexamethasone (DEX) [3]. Further tests rely on the increased Cross-Linked Actin Networks (CLANs) expressed from TM cells exposed to TGFB2 or glucocorticoids [13]. CLANs are dome-like, polygonal rearrangements in the actin cytoskeleton and are prevalent in glaucomatous TM cells [14]. It is discussed that CLAN development contributes to TM stiffness and that greater expression increases outflow resistance [14, 15]. Functional testing of TM cells includes demonstrating phagocytic properties by adding material like microbeads or latex microspheres to the medium [11, 16]. Because these features are not found in adjacent cells, these markers can be combined to identify TM cells [3, 11].

To our knowledge, only three publications reported extracting TM cells from mice. In 1991, Begley et al. suggested a method for culturing tissue samples from the anterior part of the eye. Based on morphological criteria, outgrowing cells were separated into keratocytes, corneal endothelial cells, and TM cells. All cells except TM cells were removed with a cotton swab. For further characterization, ICC was performed for laminin, COL4, and FN1 [16]. The method is limited in terms of the visual identification of cells, and the markers chosen for identification seem inadequate by current standards. A technically demanding method for dissecting a mouse eye to isolate TM cells was described by Tamm et al. in 1999, in which the murine TM strip was separated from the cornea and subsequently cultured. Cells were positive in ICC for collagen types I, III, IV, and VI, laminin, FN1, and neural cell adhesion molecule. Most

cells were negative for ACTA2, but when exposed to TGFB2, most cells expressed ACTA2 [17]. The method presented is quite complex, and, according to our preliminary tests, it turns out to be very challenging to separate the murine trabecular meshwork strip from the cornea. Only one study on gaining murine TM cells fulfills modern criteria for characterizing the extracted cells. That method involves injecting magnetic microbeads in the anterior chamber of living mice with subsequent isolation seven days later. However, this method is complex, requires experiments on living animals, and has a low yield, thus requiring many animals [11]. Therefore, we sought a cost-effective, reproducible, and animal-reducing strategy for extracting mouse TM cells.

## Materials and methods

### Materials

Chemicals, mouse strains, equipment, software and order numbers used are listed in Table 1.

### Animals

This experiment used 24 eyes of 12 one- to nine-month-old male and female C57BL/6J mice in equal numbers, and 6 eyes from four-month-old female FVB/N mice. They were obtained from our animal breeding facility (ZTL MHH, Hannover, Germany). Animal housing and studies were performed according to the German Law on Animal Welfare, Animal Welfare Regulations for Laboratory Animals, Laboratory Animal Register Law, the Directive 2010/63/EU of the European Parliament, and of the Council of 22nd September 2010 on the protection of animals used for scientific purposes. They adhered to the ARVO Statement for the Use of Animals in Ophthalmic and Vision Research. Before sacrificing, animals were kept in Euro-standard Type II cages with a filter hood. Food and water were provided ad libitum. The mice were sacrificed by cervical dislocation without prior anesthesia and/or analgesia in order to minimize any potential suffering or distress to the animals.

### TM/cornea stripe isolation

After cervical dislocation, the eyes were enucleated using pressure from curved forceps. Scissors were placed around the elevated eyes, and the optic nerve was cut. The enucleated eyeballs were dipped in 70% ethanol for 30 seconds and placed in a Petri dish filled with HBSS. Preparation took place under a dissecting microscope with adjustable magnification. First, connective tissue, fat, and conjunctiva were removed using scissors. The eyes were cut in half anterior-to-posteriorly with a scalpel and scissors, and the vitreous and the lens were removed. Subsequently, the posterior part of the eyeball was discarded by cutting it near the equator. A sterile needle was inserted through the cornea to fixate the tissue on the Petri dish. Small radiant incisions in the sclera and cornea were performed to flatten the tissue. The iris and attached ciliary body were carefully pulled away using forceps. The outflow tissue could be easily distinguished against the transparent cornea and whitish sclera as a pigmented circumferential strip. Next, the pigmented TM stripe and parts of the cornea were isolated without damaging the outflow tissue using a scalpel. The dissected cornea/TM stripe was put into a 1.5 ml Eppendorf tube containing HBSS, and the same procedure was also performed on the other half of the eyeball.

### TM/cornea stripe cultivation and TM cell expansion

Two to four isolated cornea/TM strips were transferred around the center of the well of a 6-well tissue culture plate with the pigmented side down. Excess HBSS around the cornea/TM

**Table 1. Materials.**

| Materials | Sources | Identifiers |
| --- | --- | --- |
| **Chemicals** | | |
| Accutase | Millipore | SCR005 |
| AlexaFluor™488 Phalloidin | Invitrogen | #A12379 |
| AlexaFluor™488 Goat anti-Mouse | Invitrogen | #A-11029 |
| AlexaFluor™488 Goat anti-Rabbit | Invitrogen | #A-11008 |
| AlexaFluor™546 Goat anti-Mouse | Invitrogen | #A-11003 |
| AlexaFluor™Plus 555 Phalloidin | Invitrogen | #A30106 |
| Alpha-Smooth Muscle Actin Rabbit mAb | Cell Signaling | #19245S |
| Bovine Serum Albumin (BSA) | Carl Roth | #T844.2 |
| Collagen IV Rabbit pAb | Abcam | #ab6586 |
| Dexamethasone (DEX) | Sigma-Aldrich | D4902-25mg |
| Dulbecco's Phosphate Buffered Saline (PBS) | Lonza | #BE17-512F |
| Eagle's Minimum Essential Medium (MEM) | Sigma-Aldrich | M8042-500ML |
| Ethanol | Carl Roth | #5054.1 |
| Fetal Bovine Serum (FBS) | Pan-Biotech | #P40-37500 |
| Fibronectin Mouse mAb | Invitrogen | #11324553 |
| Fluorescent Yellow Particles | Spherotech | #FL-2052-2 |
| Gelatin from cold water fish skin | Sigma-Aldrich | #G7765-250ML |
| GlutaMAX™ | Gibco | #3505–061 |
| Goat Serum | Millipore | #S26-100 mL |
| Hank's Balanced Salt Solution (HBSS) | Biowest | #L0605 |
| iScript™ Advanced cDNA Synthesis Kit | Bio-Rad | #172–5038 |
| Melanin, BioReagent, suitable for cell culture | Sigma-Aldrich | M014-100mg |
| Myocilin Mouse mAb | Sigma-Aldrich | MABN866 |
| Penicillin-Streptomycin (Pen-Strep) | Gibco | #15140–122 |
| Quick-RNA™ MicroPrep Kit | Zymo Research | #R1051 |
| Recombinant Human TGFB2 | Peprotech | #100-35B |
| Rhodamin-Phallodin | Invitrogen | #R415 |
| Roti® Histofix 4% | Carl Roth | #P087.4 |
| Roti®-Mount FluorCare DAPI | Carl Roth | #HP20.1 |
| SsoAdvanced™ Universal SYBR® Green Supermix | Bio-Rad | #1725275 |
| Triton™ X-100 | Sigma-Aldrich | #X100-100ML |
| TrypLE™ Express Enzyme | Gibo | #12604–021 |
| Tween® 20 | Sigma-Aldrich | #P9416-50ML |
| Vimentin Rabbit mAb | Cell Signaling | #5741S |
| **Animals** | | |
| C57BL/6J mice | ZTL MHH | |
| FVB/N mice | ZTL MHH | |
| **Instruments / Software** | | |
| BioTek Lionheart™ FX Automated Microscope | Agilent | www.agilent.com |
| CFX96 Touch™ Real-Time PCR Cycler | Bio-Rad | www.bio-rad.com |
| CFX Maestro™ Software | Bio-Rad | www.bio-rad.com |
| Gen5 Image Prime 3.05 | Agilent | www.agilent.com |
| GraphPad Prism 5 | GraphPad | www.graphpad.com |
| IBM SPSS Statistics 28 | IBM | www.ibm.com |
| Microscope Cover Glasses | Glaswarenfabrik Karl Hecht | #41001113 |
| Microsoft Excel 2019 | Microsoft Corporation | www.microsoft.com |

*(Continued)*

**Table 1.** (Continued)

| Materials | Sources | Identifiers |
|---|---|---|
| Routine Stereo Microscope | Leica | M80 |
| Spark® Multimode Microplate Reader | Tecan | www.tecan.com |
| Zeiss Axio Observer Microscope | Carl Zeiss | Z1 |
| ZEN-Blue Microscopy Software | Carl Zeiss | www.zeiss.com |

strip was removed with a 200 μl pipette and air-dried for 3 to 5 minutes at room temperature (RT). TM cell nutrient medium was then carefully pipetted dropwise into the 6-well until the cornea/TM strips were covered with medium but could not detach due to the meniscus effect. MEM supplemented with 1% GlutaMAX™, 10% FBS, and Pen-Strep was used as the TM cell growth medium. The cornea/TM strips were carefully transferred to a $CO_2$ incubator and cultured at 37˚C for 3–4 days to give the cornea/TM strips sufficient time to attach adequately to the surface. Next, a medium change was performed with 2 ml of culture medium. At about this time, an outgrowth of pigmented phagocytizing cells could be observed, and ½ medium change was performed twice each week. After the cells occupied approximately 20–30% of the well area, which usually occurred within the first week after the onset of cell proliferation, the cornea/TM strips were carefully removed with forceps to prevent an outgrowth of non-pigmented cell types. Since the cells primarily divide at the edges of the cell layer, it could take 1–2 weeks to reach 90% confluence. Cells were optionally split within the 6-well with Accutase to shorten this process. Therefore, the cell layer was washed twice with 2 ml of PBS, added with 300 μl of Accutase pre-warmed to 37˚C, and incubated at 37˚C for 5 minutes to separate the cell layer into single cells. Subsequently, 2 ml of culture medium was added to the cells. The cells reached 90–100% confluence through this intermediate step within 2–3 days. For the initial cell expansion, the cells were split 1:1 with 300 μl TrypLE, i.e., the cells were divided from one 6-well into two new 6-wells. For further cell expansion, cells were split in a ratio of 1:3.

## ICC staining

For the ICC staining, $5x10^4$ cells were seeded in the complete growing medium on 13-mm-diameter round microscope glass cover slides in each well of a 24-well plate. After 48 h, cells were washed twice with PBS and immediately fixated with 4% Roti® Histofix for 1 h at RT. After washing twice with PBS for 10 min at RT, cells were blocked at RT for 1 h. The blocking solution contained 2% goat serum, 1% BSA, 0,1% cold water fish skin gelatin, 0.1% Triton™X-100, and 0.05% Tween® 20 dissolved in PBS. Cells were incubated with ACTA2 rabbit mAb, VIM rabbit mAb, FN1 mouse mAb, and COL4 rabbit pAb diluted 1:1000 in the blocking solution overnight at 4˚C. After washing three times with 500 μl PBS for 10 min at RT, cells were incubated with AlexaFluor488 anti-rabbit or AlexaFluor488 anti-mouse diluted 1:1000 in PBS for two hours at RT. After 2 times washing with PBS for 10 min at RT, coverslips were mounted on a microscope slide with Roti®-Mount FluorCare DAPI. Images were obtained with an Observer Z.1 microscope equipped with an Apotome 2 using the ZEN-Blue analysis software.

## TGFB2 exposure and CLAN analysis

For TGFB2 exposure, $1x10^4$ cells were seeded on glass coverslips in each well of a 24-well plate in 2% FBS-containing growing medium together without or with 20 ng/ml TGFB2 for nine days, with media exchange every three days. After nine days, cells were fixated and blocked, as described above. After blocking, the F-Actin cytoskeleton was stained with Rhodamin-

Phalloidin diluted 1:250 in PBS for 2 h at RT. After washing with PBS for 10 min at RT twice, coverslips were mounted on a microscope slide. Images were obtained with the microscope using a 40-fold oil objective.

Cells with at least three CLANs were counted as CLAN-positive cells. Ten coverslips were analyzed, and for each coverslip, five regions were investigated, leading to 50 analyzed regions for each treatment. In those regions, the ratio of CLAN-positive cells to CLAN-negative cells was evaluated [11].

## DEX exposure

For Myoc expression, cells were treated with 500 nM DEX every three days for one week. After treatment, cells were washed twice with PBS and immediately fixated and blocked as described above. Cells were incubated with Myoc Mouse mAb diluted 1:1000 in blocking solution overnight at 4˚C. Next, cells were washed twice with PBS containing 0.1% Tween® 20 for 10 min. Afterward, the cells were incubated 1:1000 with AlexaFluor™488 phalloidin and 1:1000 Alexa-Fluor™546 goat anti-mouse for 2 h at RT. After washing thrice with PBS, coverslips were mounted on a microscope slide. Images were obtained with the microscope.

## Identification of phagocytosing cells with fluorescent microbeads or melanin

$5 \times 10^4$ TM cells were seeded on microscope glass cover slides into each well of a 24-well plate. 24 h later, 1 µl Fluorescent Yellow Particles were added to each well. After 48 h, the medium was removed, and cells were washed four times with PBS and subsequently fixated for immunostaining and fluorescent imaging with an inverted microscope as previously described, except that 1:2000 AlexaFluor™Plus 555 phalloidin was used.

The percentage of fluorescent microbeads-phagocytosed cells was also evaluated using Bio-Tek Lionheart™ FX automated microscope and Gen5 Image Prime 3.05 software. A 4x4 montage image of DAPI, green (GFP, for detecting fluorescent microbeads), and red (RFP, for detecting F-actin) channels under a 10X objective was taken in each cover slide. The 16 images were stitched into one single image and then processed for analysis. In the DAPI channel, DAPI-stained nuclei were gated as primary masks for total cell counting. Here, an object threshold over 5000 and a size between 7 to 100 µm were applied. The primary masks were expanded in the GFP channel, with 15 µm as secondary masks. Among the secondary masks, objects with a peak GFP signal over 2000 were defined as GFP-positive and counted. The percentage of fluorescent microbeads-phagocytosed cells was determined by the number of GFP-positive cells divided by the number of DAPI-stained nuclei. The data were calculated from three biological replicates and two technical replicates. For testing the phagocytotic properties of albino TM cells, either fluorescent microbeads as previously described were added, or cells were exposed to 1µg/ml melanin for 48 h. After 48 h and before image acquisition, the TM cells were washed 3 times with PBS before adding fresh media.

## Real-time PCR

For Myoc mRNA quantification, we performed real-time PCR (qPCR). Prior cells were exposed to 500 nM DEX every three days for one week. For cell lysis and RNA isolation, the Quick-RNA™ MicroPrep Kit complied with the manufacturer's protocol was used. Subsequently, the RNA concentration was measured using the Spark® plate reader nucleic acid quantification. 0.5–1 µg total RNA of each sample was reverse-transcribed with the iScript™ Advanced cDNA Synthesis Kit for RT-qPCR and further diluted to 3 ng/µl.

qPCR was performed with 2 μl cDNA, 500 nmol RT-Primer each, and 4 μl of SsoAdvanced™ Universal SYBR® Green Supermix using a CFX96 Touch™ Real-Time PCR Cycler. DNA denaturation and polymerase activation were performed for 3 min at 95˚C followed by 40 cycles (95˚C for 10 s / 55˚C for 30 s). After each cycle, SYBR green fluorescence was measured. A melt curve analysis was conducted from 65˚C to 95˚C in 0.5˚C increments to confirm the specificity of primer pairs. The used primer sequences were ACCCAGGAGCAAAGAAGGAGAC (mRT-MYOC-fwd), CAATCCTCCATGTGCTTTCCTGG (mRT-MYOC-rev), CATCACTGCCA CCCAGAAGACTG (mRT-GAPDH-fwd) and ATGCCAGTGAGCTTCCCGTTCAG (mRT-GAPDH-rev). The CFX Maestro™ Software was used for data acquisition and analysis. Samples with Ct values > 30 were not considered for further analysis. mRNA copies were normalized to the untreated control value. Three biological replicates were conducted, and four technical duplicates were averaged for each experimental setup.

### Statistic analysis

IBM SPSS Statistics 28, GraphPad Prism 5, and Microsoft Excel 2019 were used for the statistical analysis. Each data is expressed as mean ± Standard Deviation (SD). Analysis was performed with unpaired Student t-tests. A p-value of < 0.05 was considered statistically significant.

## Results

### TM/cornea stripe isolation and cultivation

Preliminary tests showed that gripping the TM with tweezers, as commonly performed in humans [3], or the method described by Tamm for murine trabecular meshwork isolation did not work for us to isolate the TM cells in mice because it is so tiny and fragile. The method introduced in this study (Fig 1) has demonstrated reproducibility, enabling the establishment of TM cell cultures from each eye separately.

The tissue strips were placed in a 6-well plate with the pigmented TM facing down. The arrangement consisted of two strips placed opposite each other or four strips arranged in a square around the center of the well (Fig 2A). This arrangement allows the meniscus effect to prevent the strips from floating.

Next, the tissue strips were air-dried for three to five minutes (Fig 2B), and the medium was added dropwise until the strips were barely covered (Fig 2C). After 3–4 days, the tissue strips were adherent enough to change medium with 2 ml medium with ½ media change twice weekly (Fig 2D and 2E). An outgrowth of pigmented, phagocytizing cells from the dissected

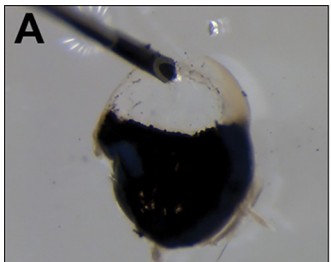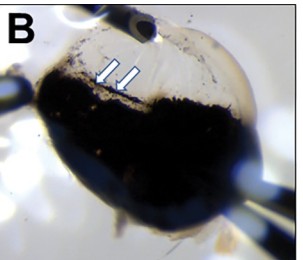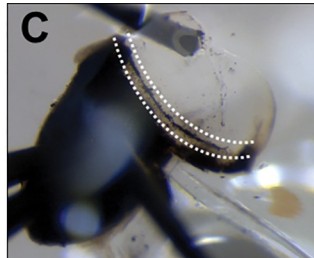

**Fig 1. Trabecular meshwork dissection.** The cornea was fixated on the plate with a needle for easier preparation. The eye was cut in half, and the lens and vitreous were removed (A). After removing the posterior part, the iris was gently grabbed and pulled backward (B). TM could be easily distinguished as a pigmented stripe (arrows). Cuts near the TM (highlighted with dotted lines) were made to isolate the TM/cornea stripe (C).

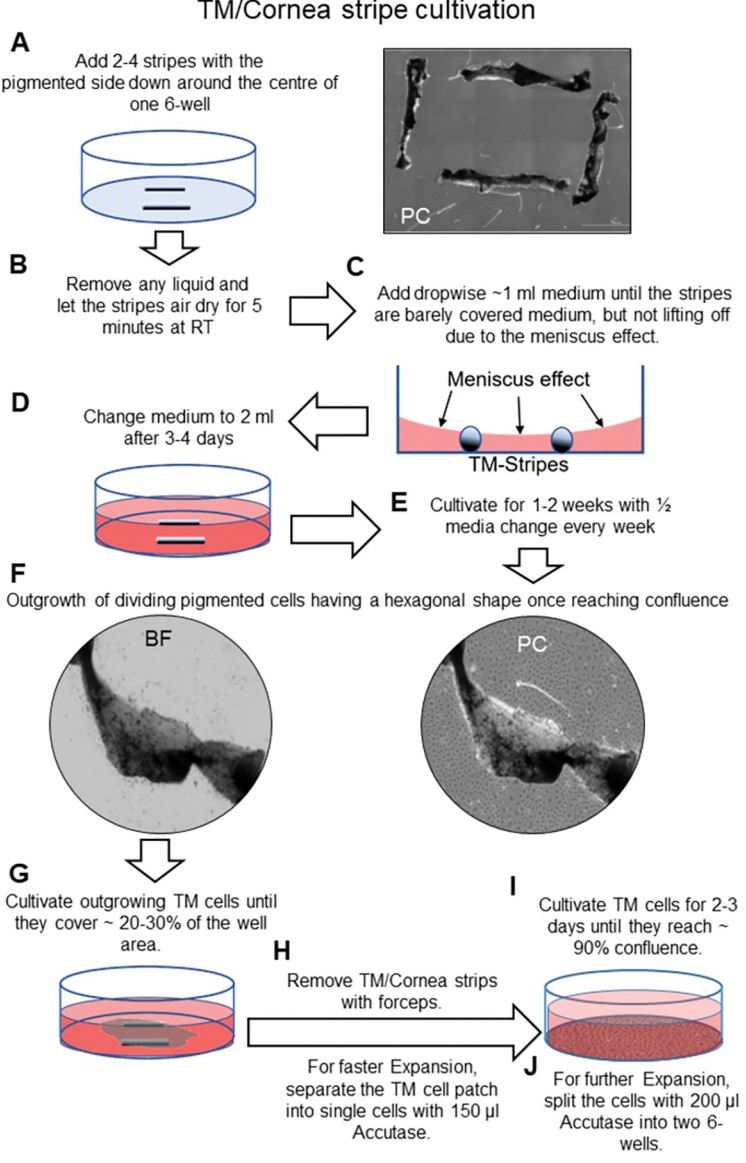

**Fig 2. Walk-through scheme of cornea/TM stipe cultivation and TM cell expansion.**

tissue band could be observed 4–10 days after the initial culture (Fig 2F and S1 Video). Their morphology resembles that of TM cells with dendritic processes, and they arrange themselves into a cobblestone-like pattern above a specific cell density. After the outgrowing cells reached 20–30% of the well area, the tissue strips were discarded with forceps (Fig 2G and 2H). Since the cells only proliferate at the edges of the cell patch, the cells were optionally separated with Accutase within the well to increase the expansion. The cells were initially expanded in a ratio of 1:1 and from passenger number 2 onwards in a ratio of 1:3 as soon as they had reached a confluence of about 90%. Therefore, one TM cell culture was initiated from each mouse eye. The cells were expanded from one 6-well plate to two 6-well plates (passage 1) and then to a T25 Falcon flask (passage 2). Finally, cells were transferred to a T75 flask culture at the third passage.

In a few cases where the TM/cornea stripe was not removed within the recommended time, an outgrowth of morphologically different, non-pigmented cells could be observed. By adding fluorescent microbeads to these cultures, it was evident that only one of the cell types possessed phagocytic properties (S1 Fig). These cultures of cells were discarded and were not used for further characterization. However, these non-wanted "cell contaminations" underline the importance of removing the cornea/TM strips in time.

TM cells changed appearance after 6 to 8 passages, and no culture reached passage nine without major senescence features like vacuoles, increased size, and highly reduced doubling time. Therefore, only cultures up to passage 5 were used for further experiments.

## TM ICC staining

To date, there is no unique marker to identify TM cells. Therefore, the expression of several markers reported previously for TM cells was validated by ICC staining, including extracellular matrix coding genes like COL4 and FN1 and VIM and ACTA2, two cytoskeleton-associated genes, to verify the identity of the cells (Fig 3).

The cells were positive for extracellular matrix proteins FN1 and COL4. VIM was widely distributed within the cytoplasm. ACTA2, also known as alpha-smooth muscle actin (αSMA), exhibited a filamentous-oriented distribution within the cytoplasm.

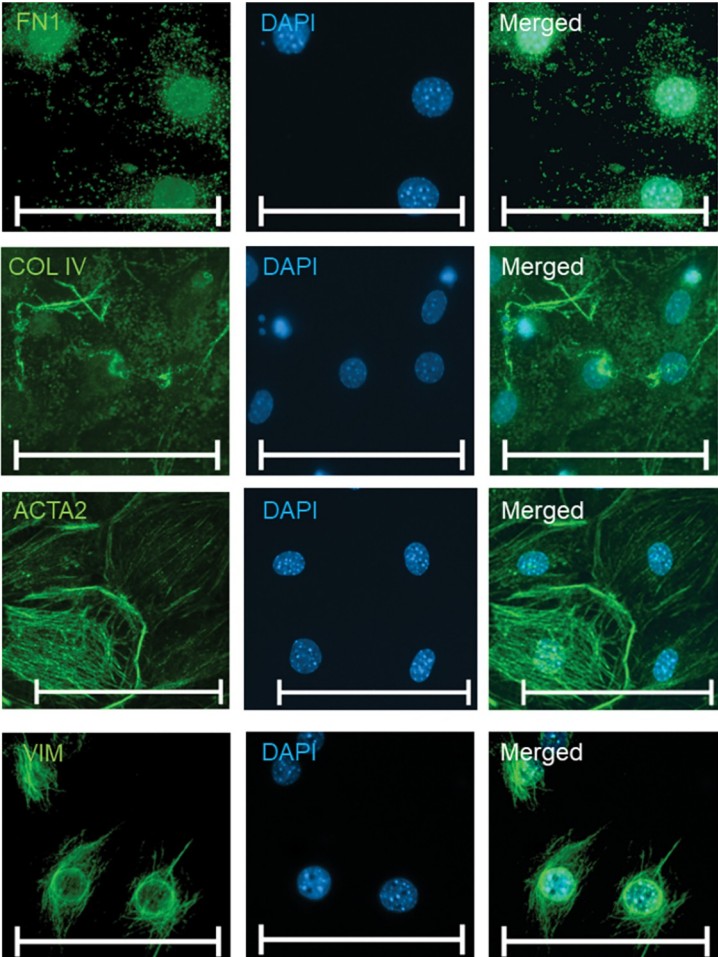

**Fig 3. ICC staining of murine TM cells.** ICC for FN1, COL4, ACTA2, and VIM are depicted in green. Cell nuclei were stained with DAPI (blue). The scale bar corresponds to 100 μm.

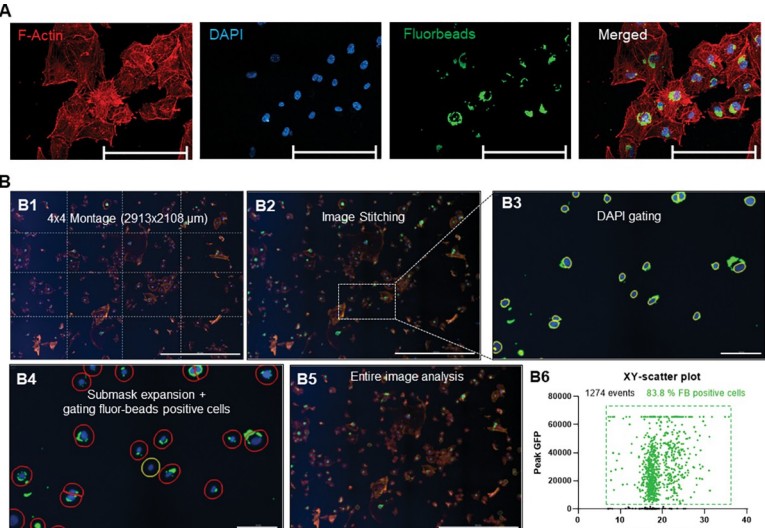

**Fig 4. Identification of phagocytosing cells with fluorescent microbeads.** (A) Representative fluorescence images of TM cells previously cultured with green fluorescent microbeads for 24 h. Nuclei were stained with DAPI and F-actin with Alexa555-Phallodin. Scale bar represents 200 μm. (B) Quantitative determination of phagocytosing cells using a live-cell imager. A montage of 4x4 images was taken for each coverslip with a 10-fold objective (B1). After image stitching (B2), the DAPI-stained nuclei were gated in the blue channel (B3). This primary mask was expanded by 15 μm and used as a sub-mask in the green channel to identify fluorobeads-positive cells (B4-5). Fluorobeads-positive cells are highlighted with red, and Fluorobeads-negative cells are highlighted with yellow circles. XY-scatter plot showing the peak GFP intensity and nuclei size of 1274 analyzed cells (B6). The fluorobeads-positive counts are highlighted in green. The scale bars corresponds 1000 μm for B1, B2 and B5, and 100 μm for B3 and B4.

## Testing phagocytic properties with fluorescent microbeads

The cells initially grown out of the cornea/TM strip could phagocytose pigment released by the preparation and showed a high degree of pigmentation (S1 Video). However, with increasing cell division, the cells lost pigmentation until they were no longer pigmented. The phagocytic properties can be regarded as a unique feature of TM cells compared to neighboring cells and thus represent a reliable marker. Therefore, the phagocytic properties of the TM cells were assessed. Cells were seeded on coverslips, treated with fluorescent microbeads for 24 h, and washed rigorously to remove non-phagocytosed beads.

The TM cells could phagocytize the beads, confirmed by fluorescence staining using Alexa555-Phalloidin and the nuclear stain DAPI. They showed an accumulation of a varying number of green fluorescent beads around the nucleus in the cytoplasm (Fig 4A). A live-cell imager was used to record a 4x4 montage per coverslip with a 10-fold objective for further quantification. The 16 individual images were stitched together into one image, and the cell nuclei in the DAPI channel were gated with a primary mask. The primary mask was expanded by 15 μm and used as a sub-mask in the green channel to identify cells with fluorescent beads (Fig 4B). Three biological replicates in two technical replicates were analyzed. Thus, 1274 cells were analyzed, of which 83.8% of the cells had phagocytosed fluorobeads within 24 h (Fig 4B6).

## Isolation of non-pigmented TM cells from albino mice

Additionally, TM cells from albino mice strain FVB/N were isolated to demonstrate that the present approach can obtain TM cells from other mouse strains. The difficulty here is that in albino mice, the TM lacks pigmentation and does not stand out visually from the cornea, making manual isolation, as previously described by Tamm et al., impossible. Therefore, a stripe

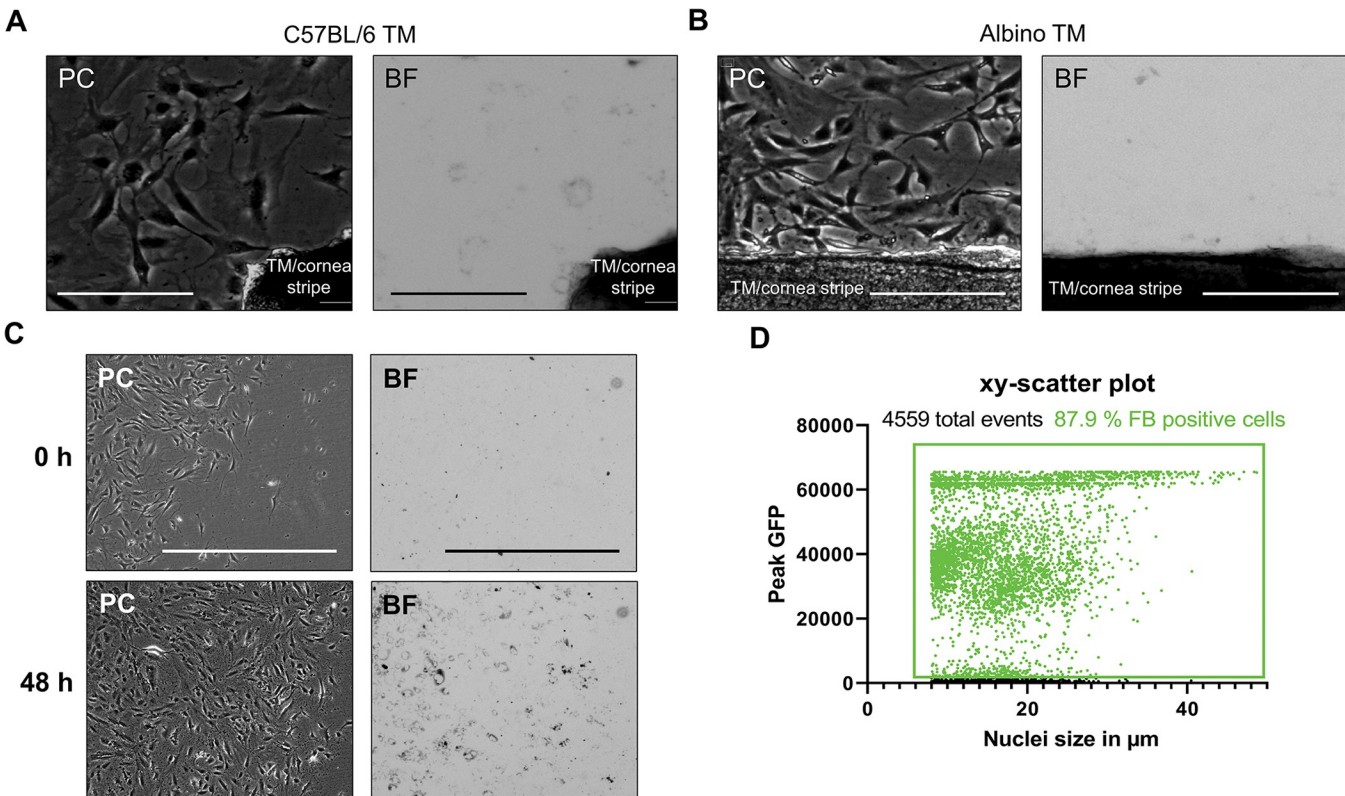

**Fig 5. Isolation of non-pigmented TM cells from albino mice with phagocytotic properties.** (A) Representative phase-contrast (PC) and bright-field images (BF) of TM cells growing out of the TM-cornea stripe isolated from C57BL/6 mice 6 days after dissection. The scale bar represents 150 μm. (B) Representative phase-contrast and bright-field images of TM cells growing out of the TM-cornea stripe isolated from albino mice 5 days after dissection. The scale bar represents 150 μm. (C) Representative phase-contrast and bright-field images from passage 2 albino TM cells were exposed to 1μg/ml melanin for 0 h(upper row) and 48 h (lower row). The scale bar represents 1000 μm. (D) XY-scatter plot showing the peak GFP intensity and nuclei size of 4559 analyzed albino TM cells exposed to fluorobeads for 24 h. The fluorobeads-positive cells are highlighted in green.

along the border between the cornea and sclera was cut out and cultivated as described above. 4–6 days after initial dissection, the first cells were growing out of the TM/cornea stripe (SV2). In contrast to the cells isolated from C57BL/6 mice, these cells did not show any degree of pigmentation (Fig 5A and 5B). To confirm the phagocytotic properties, albino TM cells were seeded and exposed to 1 μg/ml melanin for 48 hours (Fig 5C). An accumulation of melanin could be observed in the albino TM cells. For quantification of phagocytic cells, the experiment with fluorobeads was also carried out with albino TM cells, as described above. 24 h after fluorobead exposure, 4559 cells were analyzed, of which 87.9% had successfully taken up fluorobeads (Fig 5D).

## TM response after TGFB2 or DEX exposure

TM cells react in a specific way after being exposed to TGFB2 and DEX. Therefore, the TGFB2-induced expression of CLANs and the increased expression of Myoc after DEX exposure were assessed. CLANs are polygonal rearrangements in the actin cytoskeleton to respond to stress like TGFB2 or DEX. However, they can also be found in a considerably smaller proportion during cell attachment and spreading phases [13]. The cultures treated with 20 ng/ml TGFB2 for 9 days increased their expression of CLANs (Fig 6A and 6B). In the control group, 12.91% (n = 10; SD: 9.68%) of the DAPI-positive cells showed an expression of CLANs. In the

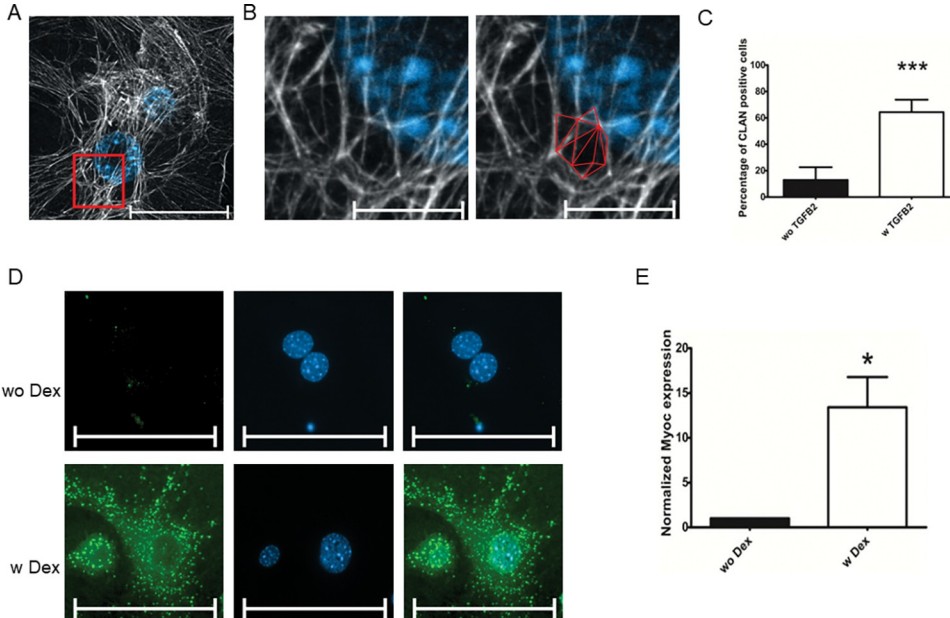

**Fig 6. TM response after TGFB2/DEX exposure.** (A) CLANs are polygonal rearrangements in the actin cytoskeleton expressed by TM cells after TGFB2 exposure. ICC for F-Actin (white) and nuclei were stained with DAPI (blue). The scale bar corresponds to 50 μm. (B) Enlarged view of the area in the red rectangle from (A). The triangles forming the dome-like CLANs have been marked in red. The scale bar corresponds to 10 μm. (C) Percentage of CLAN-positive cells treated without or with 20 ng/ml TGFB2 for nine days. The data show the results of 5 biological replicates performed in two technical replicates each. For each replicate, five regions per coverslip were analyzed. A paired t-test (***p < 0.0001) was performed to assess the significant increase in CLANs expression by TGFB2. (D) ICC staining of TM cells for Myoc (green) and cell nuclei were stained with DAPI. Non-treated cells show little to no Myoc expression, but after treatment with 500 nM DEX every three days for one week, TM cells show high Myoc expression. The scale bar corresponds to 100 μm. (E) qPCR analysis of Myoc without or under DEX treatment. Myoc expression was normalized to GAPDH and further normalized to the non-treated control. RT-Data was performed in 3 biological replicates with 4 technical replicates. A one-sample t-test confirmed the significance of the increase (*p = 0.024, n = 3).

treated group, 64.32% (n = 10; SD: 9.46%) of the DAPI-positive cells showed an expression of CLANs. A paired t-test confirmed the significance of the increase (p < 0.0001, Fig 6C). Finally, Myoc expression was analyzed, a protein increasingly expressed in response to DEX [3]. In ICC staining, non-treated cells showed subtle expression of Myoc, and cells treated with 500 nM DEX every three days for one week showed increased expression of Myoc (Fig 6D). The same settings, including histogram and exposure time, were used for this comparison in ICC. This significant increase in the expression of Myoc upon DEX treatment was also verified in qPCR analysis performed with cultures of different passages (Fig 6E).

## Discussion

In this study, we present a novel method of TM cell extraction in mice. The cells isolated in our experiments showed typical TM cell characteristics, including expression of COL4, FN1, VIM, and ACTA2 and an increased expression of Myoc after DEX stimulation. They increased the expression of CLANs after TGFB2 treatment [3, 11, 18]. Specific phagocytic properties were also observed in the isolated cells [3]. Therefore, our method is highly suitable for extracting TM cells.

A problem with isolating and characterizing TM cell cultures lies in the anatomy of the TM. The human TM is divided into the uveal, the corneoscleral, and the juxtacanalicular meshwork [12]. No markers have been reported to distinguish all three sections [3]. It is still controversial

whether the different sections of the TM are made up of different cell types or whether one cell type behaves differently under various conditions [19]. The anatomy of the TM in mice varies slightly. The TM outflow pathway is separated into an inner component that resembles the corneoscleral TM in humans and consists of one or two connective lamellae covered by flat cells and an outer layer. The outer layer lacks structured lamellae and resembles a loose connective tissue akin to the juxtacanalicular TM in humans [3]. TM cells share one property in humans and mice: their ability to phagocytosis [11, 20].

The dissection technique itself posed another limitation to this study. It relies on the visibility of the outflow pathway as a pigmented circumferential band, which is absent in albino mice [21]. However, it is essential to highlight that the method introduced here has proven successful even in the case of albino animals. This achievement holds significant importance, considering that the absence of orientation markers in such animals presents challenges for dissection techniques [3]. Overcoming this obstacle is noteworthy, as it extends the range of research possibilities by potentially allowing for the application of other mouse strains. However, inaccurate preparation due to lower visibility of TM or too late removal of the cornea/ TM stripe may result in a mixed culture contaminated by other cell types. Here, adding fluorescent or magnetic microspheres followed by magnetic cell separation or fluorescence-activated cell sorting (FACS) could provide the opportunity to isolate TM cells from mixed cell cultures [22]. Combining our presented cornea/TM stripe cultivation with magnetic cell separation or FACS could help transfer this method to other species with smaller eyes and more complex preparation settings, such as zebrafish, also used in glaucoma research [23].

So far, to the authors' knowledge, there have been only three published reports on dissecting techniques for extracting TM cells in mice. In 1991, Begley and colleagues proposed a technique that involved culturing tissue samples from the anterior part of the eye and classifying outgrowing cells into three types based on morphological features: keratocytes, corneal endothelial cells, and TM cells [16]. However, this method has certain limitations regarding the optical identification of cells, and the markers used for identification are insufficient by modern standards. Another method was published in 1999 by Tamm and colleagues, which is a time-consuming and technically demanding procedure [3, 17]. It involves displaying and dissecting the zonule in mice under 40x magnification, inserting a forceps in the suprachoroidal space, and cutting the attachments of the ciliary body with a fine knife while grasping it [17]. Also, due to its publication date, it did not meet modern criteria for characterizing cells and did not investigate Myoc induction by DEX [3]. However, our presented method does not require such intricate steps as used by Tamm et al. By eliminating the need for visualizing and dissecting the zonule and by choosing to grasp the iris together with the attached ciliary body instead of maneuvering a forceps into the suprachoroidal space while simultaneously using a knife to release adhesions with the other hand, the method introduced here presents a heightened level of simplicity. It is, therefore, much easier to reproduce. Additionally, a broad range of TM cell markers was investigated to verify their identity.

A modern method for TM cell isolation uses the phagocytic feature of TM cells and magnetic beads for cell isolation [11]. However, this is a costly method that requires injecting beads into living animals. There, tissue from up to 15 mice is pooled for one cell culture, requiring many animals. An approach where cells are acquired post-animal euthanasia offers benefits compared to a procedure involving injections into live animals followed by their euthanasia for cell retrieval. Moreover, it is worth noting that in numerous countries, obtaining approvals for injecting substances into live animals for research tends to involve more intricate bureaucratic procedures than regulations surrounding euthanasia for organ harvesting. Therefore, an inexpensive and animal experiment-reducing strategy for isolating murine

TM cells is required. We suggest that our study provides this inexpensive and animal-reducing strategy.

One strength of the method described here is the yield. Depending on whether two or four TM/Cornea strips are used for the initial cell culture, one to two TM cell cultures per mouse can be obtained. Already from passage number 3 a confluent T75 flask can be achieved. The reduction of the required animals underlines the animal-reducing aspect of the presented method.

The improved availability of TM cells due to this easily reproducible method generates several exciting possibilities. Many translational approaches in mouse glaucoma research aim to influence the TM to increase outflow capacity [9]. Because of the improved availability of TM cells, many experiments could be done primarily *in vitro* and strengthen animal welfare by the replace-reduce-refine method for experiments on living animals. Additionally, the characteristics of isolated TM cells from different mouse strains, like knock-out models, could be investigated *in vitro*. These options open a new field of research for glaucoma research.

## Supporting information

**S1 Video. Time-lapse video of proliferative, phagocytosing, and pigmented cells growing out of the cornea/TM strip one week after preparation.** Phase contrast images are shown on the left, and bright field images on the right.
(MP4)

**S2 Video. Time-lapse video of proliferative, phagocytosing, and non-pigmented cells growing out of the cornea/TM strip previously isolated from albino mice 6 days after preparation.** Phase contrast images are shown on the left, and bright field images on the right.
(MP4)

**S1 Fig. Overview of a mixed cell culture, three weeks after initial preparation, where the cornea/TM strip was not removed in time.** The addition of fluorobeads for 48 h and subsequent washing steps reveals that only one cell layer exhibits phagocytotic properties.
(TIF)

**S1 Raw data. Raw data used to generate Figs 4B6, 5D, and 6C+6E.**
(XLSX)

## Author Contributions

**Conceptualization:** Heiko Fuchs.

**Data curation:** Maximilian Binter, Fridolin Langer, Xiaonan Hu, Migle Lindziute.

**Formal analysis:** Maximilian Binter, Xiaonan Hu, Heiko Fuchs.

**Funding acquisition:** Carsten Framme.

**Investigation:** Maximilian Binter, Fridolin Langer, Xiaonan Hu.

**Methodology:** Xiaonan Hu, Heiko Fuchs.

**Supervision:** Carsten Framme, Jan Tode, Heiko Fuchs.

**Writing – original draft:** Maximilian Binter, Heiko Fuchs.

**Writing – review & editing:** Maximilian Binter, Fridolin Langer, Xiaonan Hu, Migle Lindziute, Carsten Framme, Jan Tode, Heiko Fuchs.

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
