## [Decision Letter · Decision Letter 0]

25 Jul 2023

PONE-D-23-21644A simple and animal-reducing dissection method for the isolation of mouse trabecular meshwork cellsPLOS ONE

Dear Dr. Fuchs,

Thank you for submitting your manuscript to PLOS ONE. After careful consideration, we feel that it has merit but does not fully meet PLOS ONE’s publication criteria as it currently stands. Therefore, we invite you to submit a revised version of the manuscript that addresses the points raised during the review process.

We look forward to receiving your revised manuscript.

Kind regards,

Ashok Kumar, Ph.D.

Academic Editor

PLOS ONE

2. To comply with PLOS ONE submissions requirements, in your Methods section, please provide additional information regarding the experiments involving animals and ensure you have included details on methods of anesthesia and/or analgesia

“No, the funders had no role in study design, data collection and analysis, decision to publish, or preparation of the manuscript.”

6. We notice that your supplementary figures are uploaded with the file type 'Figure'. Please amend the file type to 'Supporting Information'. Please ensure that each Supporting Information file has a legend listed in the manuscript after the references list.

Additional Editor Comments:

You manuscript has been evaluated by three experts in the field. Please respond to each critique carefully. One of the reviewer has suggested to change the title as "animal-reducing" is not fully justified.

Reviewers' comments:

Reviewer's Responses to Questions

**Comments to the Author**

1. Is the manuscript technically sound, and do the data support the conclusions?

Reviewer #1: Yes

Reviewer #2: Yes

Reviewer #3: Partly

2. Has the statistical analysis been performed appropriately and rigorously? 

Reviewer #1: Yes

Reviewer #2: Yes

Reviewer #3: Yes

3. Have the authors made all data underlying the findings in their manuscript fully available?

Reviewer #1: Yes

Reviewer #2: Yes

Reviewer #3: Yes

4. Is the manuscript presented in an intelligible fashion and written in standard English?

Reviewer #1: Yes

Reviewer #2: Yes

Reviewer #3: Yes

5. Review Comments to the Author

Reviewer #1: The manuscript entitled “A simple and animal-reducing dissection method for the isolation of mouse trabecular meshwork cells” by Binter and coworkers has described a method for TM cell isolation from mice. This is a methodological paper that could be interesting for glaucoma researchers. TM cell extraction is very challenging from the mouse, and therefore, this paper could provide a potential protocol for TM cell isolation.

Authors criticized the existing literature on TM extraction as labor-intensive and cumbersome. The authors also criticized these published protocols for their low yield; however, it is not clear what the TM yield was in the current study. How many mice should be used for what % yield? The results demonstrated for IFA and other techniques look very low in number in the representative frame.

The authors have not provided any quantitive data regarding yield from the 24 eyes (12 animals) used in this study. Mao et al. have used 13-15 mice, and this study utilized 12 mice which are not significantly different; therefore, the title needs to be modified by removing ‘animal-reducing.’

By citing a paper by Mao et al. (where they used magnetic beads for phagocytic TM extraction), the authors stated that this could be unethical (or pose ethical dilemma); however, the microbead occlusion model, where the microbeads are injected into the mouse eyes, is a well-known model and frequently used in the field. Authors should modify this statement in their discussion.

The authors stated that removing the TM/corneal ring is crucial as it contaminates the TM culture with other cell types. If this is critical, the authors should provide the time frame for this step. On what days after dissection did the authors reach 20-30% confluency and remove their TM/Corneal ring?

Line 340-342, for % data representation, all ‘comas’ should be replaced with ‘periods’ (e.g., 12,91% to 12.91%).

Results and some figure legends need modification. The results section is redundant to the methods, and many paragraphs are copied as such. Similarly, figure legends (e.g., Fig 4 legend) are also redundant to methods/results. In the results section, authors should only describe the results in terms of their findings/observations. In Fig. legend, they should write the technique in brief.

Fig 4 bar represents 100µm, while in Fig 3, it is written as 50µm; however, both frames looks the same, authors should check this for any potential error.

Reviewer #2: Article " A simple and animal reducing dissection method for the isolation of mouse trabecular meshwork cells" claims to establish a new method which would reduce the number of animals required to obtain trabecular meshwork cells from mice.

Further, authors speculate that this method could be extended to other small model animals used in research studies. However, the claim to use less number of animals, 12 animals used in the proposed method against about 15 animals required for one of the existing dissection method, remans to be established. The number of animals used will also be dependent on the ease of dissection and the dissection skills of the researcher therefore, the term "animal-reducing" could be little misleading or a claim made too early.

The good: Article identifies an existing issue in research field and hence establishes the importance of the work neatly. Authors (a) have clearly mentioned the experiments and results, and accepted the limitations of their method in discussion section. (b) Authors stayed clear from making any over-speculations or exaggerated claim.

The not so good: The claim made as this method will reduce the number of animals required, need to be further established. Therefore, my recommendation would be to drop the term "animal-reducing" from the title.

Apart from that the work is nicely written can be published after (a) dropping the "animal-reducing" from the title and (b) minor corrections suggested below:

L45: "TGFB2": the term needs to be explained as it has been done for trabecular meshwork (TM) in L25

L46: "Myocilin" should be replaced with "myocilin"

L71: "immunocytochemistry" staining should be replace by "immunocytochemical" staining.

L74: "Myoc" should be "MYOC" to be consistent with VIM, FN1 and other proteins all capital abbreviation.

L77-78 "The contribution of CLANs.......resistance is discussed[14,15]." sounds incomplete.

L90: Replace trabecular meshwork with TM

L92: "TGBF1" is 1 a typo? if not, TGBF1 need to be explained

L135:"Two to four........around the center of a 6 well with the pigmented side down." could be written as " Two to

four....around the center of the well of a 6 well tissue culture plate with the pigmented side down."

L161: "one h" should be replaced with "1h"

A video of the dissection and isolation of trabecular meshwork strip could be helpful to establish the ease and reproducibility of dissection methods since these are part of the claims made by authors.

Reviewer #3: Please see the attached document for the detail comments. Good Luck!

Summary

In present study, authors showed dissection method to isolate mouse trabecular meshwork (TM) cells from C57BL/6J mice. Isolated cells were characterized with (i) immunocytochemistry (ICC) for the known markers Collagen IV, Fibronectin I, Vimentin (ii) myocilin (Myoc) induction by dexamethasone (DEX) with ICC & qPCR technique (iii) increased Cross-Linked Actin Networks (CLANs) upon exposure to TGFB2 via ICC, and (iv) phagocytic properties using fluorescent microbeads. Such analysis clearly demonstrated successful isolation of TM cells.

Overall, study clearly demonstrated successful isolation of TM cells from wild type C57BL/6J mice but lacks the evidence to support their claim that their method is simple, easily re-producible, inexpensive, animal reducing and high yield! I recommend it for major revision in this regard.

6. PLOS authors have the option to publish the peer review history of their article (what does this mean?). If published, this will include your full peer review and any attached files.

Reviewer #1: No

Reviewer #2: No

Reviewer #3: No

---

## [Author Response · Author response to Decision Letter 0]

1 Nov 2023

First, we thank the editor and reviewers for taking the time to read and review our manuscript. Based on their comments and suggestions, we have improved the manuscript and would like to address their points in detail below.

Responses to the Editor comments.

Reply: Thanks for this note. We changed the file naming and changed and have changed the formatting, where we noticed differences to PLOSOne_formatting.

2. To comply with PLOS ONE submissions requirements, in your Methods section, please provide additional information regarding the experiments involving animals and ensure you have included details on methods of anesthesia and/or analgesia

Reply: The animals are euthanized through cervical dislocation. The cervical dislocation was done without prior anesthesia, as prior anesthesia would increase stress for the animals. Cervical dislocation is the fastest and, therefore, the most ethically justifiable method. No experiments were conducted on living animals.

"No, the funders had no role in study design, data collection and analysis, decision to publish, or preparation of the manuscript."

b) State what role the funders took in the study. If the funders had no role in your study, please state: "The funders had no role in study design, data collection and analysis, decision to publish, or preparation of the manuscript."

d) If you did not receive any funding for this study, please state: "The authors received no specific funding for this work."

Reply: Thanks for pointing this out. The authors received no specific funding for this work. We included this statement in the cover letter.

Reply: We would like to make changes to our Data Availability statement, stating “No – some restrictions will apply”. The data that support the findings of this study are available on request from the corresponding author, [HF]. We, the authors, have included that statement in the Cover letter.

5. We note that you have included the phrase "data not shown" in your manuscript. Unfortunately, this does not meet our data sharing requirements. PLOS does not permit references to inaccessible data. We require that authors provide all relevant data within the paper, Supporting Information files, or in an acceptable, public repository. Please add a citation to support this phrase or upload the data that corresponds with these findings to a stable repository (such as Figshare or Dryad) and provide and URLs, DOIs, or accession numbers that may be used to access these data. Or, if the data are not a core part of the research being presented in your study, we ask that you remove the phrase that refers to these data.

Reply: Thanks for pointing out this mistake. We deleted the phrase.

6. We notice that your supplementary figures are uploaded with the file type 'Figure'. Please amend the file type to 'Supporting Information'. Please ensure that each Supporting Information file has a legend listed in the manuscript after the references list.

Reply: We will upload the Supplementary Figure to the file type Supporting Information . We ensured that each Supporting Information file has a legend listed in the manuscript after the references list.

Reply: We did not cite any retraced papers and did not add new references.

Responses to the Reviewer #1 comments.

Reviewer 1:

The manuscript entitled "A simple and animal-reducing dissection method for the isolation of mouse trabecular meshwork cells" by Binter and coworkers has described a method for TM cell isolation from mice. This is a methodological paper that could be interesting for glaucoma researchers. TM cell extraction is very challenging from the mouse, and therefore, this paper could provide a potential protocol for TM cell isolation.

Authors criticized the existing literature on TM extraction as labor-intensive and cumbersome. The authors also criticized these published protocols for their low yield; however, it is not clear what the TM yield was in the current study. How many mice should be used for what % yield? The results demonstrated for IFA and other techniques look very low in number in the representative frame.

Reply: The term "high yield" was intended to underscore the ability of the presented method to generate a TM cell culture from one mouse eye. We have further emphasized this point in the manuscript with the following (Page 14, line 276): "Therefore, one TM cell culture was initiated from each mouse eye. Transferring the culture from one 6-well into two 6-well plates (passage 1) to a T25 Falcon flask (passage 2), then progressing to a T75 flask culture was obtained by the third passage."

The authors have not provided any quantitive data regarding yield from the 24 eyes (12 animals) used in this study. Mao et al. have used 13-15 mice, and this study utilized 12 mice which are not significantly different; therefore, the title needs to be modified by removing 'animal-reducing.'

Reply: Thank you for your comment. We understand that "animal-reducing" needs to be dropped to avoid confusion and exaggeration. However, we would like to briefly explain why this term was initially chosen. In Mao's method, 15 animals were used and pooled to obtain one culture. In the method presented here, one culture was obtained from one to two eyes of an animal, depending on whether 2 or 4 cornea/TM strips were used for the initial culture. However, 12 animals were necessary to test the reproducibility. We made sure always to perform 3 technical and 3 biological replicates. Furthermore, cultures were only used up to passage 5 to avoid the influence of cell senescence after too many passages.

By citing a paper by Mao et al. (where they used magnetic beads for phagocytic TM extraction), the authors stated that this could be unethical (or pose ethical dilemma); however, the microbead occlusion model, where the microbeads are injected into the mouse eyes, is a well-known model and frequently used in the field. Authors should modify this statement in their discussion.

Reply: We appreciate your input on this matter. It is important to note that we acknowledge the validity of this established approach and do not intend to call it unethical. Our standpoint is that a cell harvesting method involving the euthanasia of animals prior to cell extraction is more favorable than a procedure in which experiments are conducted on living animals, followed by their euthanasia for cell harvesting. We are grateful for this observation regarding the phrasing mistake, which could have potentially led to discontent. We changed the phrase to (page 20, line 443): "An approach where cells are acquired post-animal euthanasia offers benefits compared to a procedure involving injections into live animals followed by their euthanasia for cell retrieval. Moreover, it is worth noting that in numerous countries, obtaining approvals for injecting substances into live animals for research tends to involve more intricate bureaucratic procedures than regulations surrounding euthanasia for organ harvesting."

The authors stated that removing the TM/corneal ring is crucial as it contaminates the TM culture with other cell types. If this is critical, the authors should provide the time frame for this step. On what days after dissection did the authors reach 20-30% confluency and remove their TM/Corneal ring?

Reply: Thank you very much for your comment. Certainly, the specific timing of outgrowth varies in different cultures, which led us to use the percentage of expansion as a descriptive unit. However, we inadvertently overlooked the importance of specifying a time frame for special attention to stipes expansion. Therefore, we have now included this information in the text (page 9, line 146): "After the cells occupied approximately 20-30% of the well area, which usually occurred within the first week after the onset of cell proliferation, the cornea/TM strips were carefully removed with forceps to prevent an outgrowth of non-pigmented cell types."

 [...]".Line 340-342, for % data representation, all 'comas' should be replaced with 'periods' (e.g., 12,91% to 12.91%).

Reply: Thanks for pointing out this mistake. We have changed the comas to periods.

Results and some figure legends need modification. The results section is redundant to the methods, and many paragraphs are copied as such. Similarly, figure legends (e.g., Fig 4 legend) are also redundant to methods/results. In the results section, authors should only describe the results in terms of their findings/observations. In Fig. legend, they should write the technique in brief.

Reply: We appreciate your comments. However, we disagree with the figure legends. We believe that the figure legends, including Figure 4, provide a concise overview of the quantification of phagocytosing cells. We intend that the combination of figure and legend is understandable even without reading the entire methods section, which is the prevailing reading style of publications today. We thank her for pointing out the overlap between methods and results for the record, a common challenge in method-oriented studies. Deciding what is a method and what is a result can be complicated. We have revised the manuscript with this in mind and tried to minimize redundancy.

Fig 4 bar represents 100µm, while in Fig 3, it is written as 50µm; however, both frames looks the same, authors should check this for any potential error.

Reply: We appreciate your comment. It is worth noting that in Fig. 3, we've denoted that "The scale bar represents 100 µm," while in Fig. 5, we have denoted that "The scale bar corresponds to 50 µm" only for panel A," The scale bar corresponds to 10 µm" for panel B, and "The scale bar corresponds to 100 µm" for panel D. 

We have examined and can confirm the accuracy of these statements. Regrettably, we inadvertently omitted the scale bar in Fig. 4, but we rectified this error by including it in the figure legend.

Reviewer #2

Article " A simple and animal reducing dissection method for the isolation of mouse trabecular meshwork cells" claims to establish a new method which would reduce the number of animals required to obtain trabecular meshwork cells from mice. Further, authors speculate that this method could be extended to other small model animals used in research studies. However, the claim to use less number of animals, 12 animals used in the proposed method against about 15 animals required for one of the existing dissection method, remains to be established. The number of animals used will also be dependent on the ease of dissection and the dissection skills of the researcher therefore, the term "animal-reducing" could be little misleading or a claim made too early.

The good: Article identifies an existing issue in research field and hence establishes the importance of the work neatly. Authors (a) have clearly mentioned the experiments and results, and accepted the limitations of their method in discussion section. (b) Authors stayed clear from making any over-speculations or exaggerated claim.

The not so good: The claim made as this method will reduce the number of animals required, need to be further established. Therefore, my recommendation would be to drop the term "animal-reducing" from the title.

Response: Thanks for the comment. We understand that "animal-reducing" needs to be deleted to avoid confusion and exaggeration. However, we would like to explain why this term was chosen initially briefly. In Mao's method, 15 animals are pooled into one culture. In the method presented here, one culture was obtained from one to two eyes of an animal, depending on whether 2 or 4 cornea/TM strips were used for the initial culture. However, the 12 animals were necessary to obtain enough cultures for the experiments and, most importantly, to test the reproducibility of our method. We made sure always to perform 3 technical and 3 biological replicates. In addition, the cultures were used only up to the fifth passage to avoid the influence of cell senescence after too many passages.

Apart from that the work is nicely written can be published after (a) dropping the "animal-reducing" from the title and (b) minor corrections suggested below:

L45: "TGFB2": the term needs to be explained as it has been done for trabecular meshwork (TM) in L25

L46: "Myocilin" should be replaced with "myocilin"

L71: "immunocytochemistry" staining should be replace by "immunocytochemical" staining.

L74: "Myoc" should be "MYOC" to be consistent with VIM, FN1 and other proteins all capital abbreviation.

L77-78 "The contribution of CLANs.......resistance is discussed[14,15]." sounds incomplete.

L90: Replace trabecular meshwork with TM

L92: "TGBF1" is 1 a typo? if not, TGBF1 need to be explained

L135:"Two to four........around the center of a 6 well with the pigmented side down." could be written as " Two to

four....around the center of the well of a 6 well tissue culture plate with the pigmented side down."

L161: "one h" should be replaced with "1h"

Reply: Thanks for those helpful comments. We changed the manuscript accordingly.

A video of the dissection and isolation of trabecular meshwork strip could be helpful to establish the ease and reproducibility of dissection methods since these are part of the claims made by authors.

Reply: Answer: We appreciate that feedback. A well-done video could further illustrate the method. Unfortunately, our current resources are insufficient to capture high-quality images with our binocular microscope during preparation. Our attempts in this regard have consistently resulted in partially blurred and out-of-focus videos. We believe that recording such videos would detract from the overall professionalism of the manuscript, so we decided against showing them. Nevertheless, we are confident that our combination of visuals and textual descriptions will provide clear and easy-to-understand guidance.

Reviewer 3:

Summary

In present study, authors showed dissection method to isolate mouse trabecular meshwork (TM) cells from C57BL/6J mice. Isolated cells were characterized with (i) immunocytochemistry (ICC) for the known markers Collagen IV, Fibronectin I, Vimentin (ii) myocilin (Myoc) induction by dexamethasone (DEX) with ICC & qPCR technique (iii) increased Cross-Linked Actin Networks (CLANs) upon exposure to TGFB2 via ICC, and (iv) phagocytic properties using fluorescent microbeads. Such analysis clearly demonstrated successful isolation of TM cells.

Overall, study clearly demonstrated successful isolation of TM cells from wild type C57BL/6J mice but lacks the evidence to support their claim that their method is simple, easily reproducible, inexpensive, animal reducing and high yield! I recommend it for major revision in this regard.

Major comments:

• Authors claimed that their method is simple, easily reproducible, inexpensive, animal

reducing and high yield! However, data do not support those claims and below are some

reasons to conclude it:

1) Dissection method is not very different from what is described by Tamm et. al. 1999,

so it's not justified that the method used here is anyway simpler or better! It's only

the assay demonstrating phagocytic properties of isolated TM cells which wasn't

available in earlier time.So, it is possible to do the same assay with TM cells isolated by Tamm et. al. if they have frozen stock available, then compare! This can provide some conclusive

evidence.

Reply: We highly appreciate that feedback. While the variance from Tamm et al.'s approach might not be substantial, it is worth noting that their method involves intricate mechanical procedures that could pose challenges for specific individuals (like the authors of this manuscript). Due to certain aspects of the method, reproducibility could become an issue (as experienced by the authors of this manuscript). In particular, steps such as visualizing and dissecting zonular fibers in a mouse under 40x magnification or maneuvering a forceps into the suprachoroidal space to grasp the ciliary body while simultaneously using a knife to dissect adhesions with the other hand present notable technical complexities. We have adapted this method to a somewhat less refined yet more straightforward approach. We believe this modification holds value for publication, as it underscores the simplification we have achieved as a resulting distinction from Tamm's technique. These distinctions are outlined in detail within the manuscript (Page 19, line 434):

"However, our presented method does not require such intricate steps as used by Tamm et al. By eliminating the need for visualizing and dissecting the zonule and by choosing to grasp the iris together with the attached ciliary body instead of maneuvering a forceps into the suprachoroidal space while simultaneously using a knife to release adhesions with the other hand, the method introduced here presents a heightened level of simplicity. It is, therefore, much easier to reproduce."

We believe that comparing the phagocytic properties of cells isolated through our method and those isolated using Tamm's method would not contribute significantly to the manuscript. Our emphasis lies in presenting a reproducible isolation method and confirming the identity of these cells as TM cells.

However, we also followed your suggestion and tested our method on C57/BL/6J mice and three animals of the albino mouse strain FVB/N (Fig. 5 and SV2). In these albino animals, the trabecular meshwork is not pigmented, making the method of Tamm, which is based on the optical recognition of the pigmented TM stripe, not applicable. Our method allows us to obtain non-pigmented TM cells from albino mice. The phagocytotic properties were evaluated by adding melanin, and the percentage of phagocytosing cells was quantified by adding fluorobeads.

2) There were 12 animals (24 eyes) used, and they were wild-type C57BL/6J mice. No

quantification data showing the total yield of TM cells obtained. It is important to

prove the claim of "high-yield" method. Also, missing control to even compare the

qualitative properties!

Reply: Thanks for this valuable input, which we sincerely appreciate. The term "high yield" was deliberately chosen to highlight the capacity of the presented technique in effectively cultivating a TM cell culture from a single mouse eye. To underscore this point, we have reinforced it within the manuscript (Page 14, line:276): "Therefore, one TM cell culture was initiated from each mouse eye. Transferring the culture from one 6-well into two 6-well plates (passage 1) to a T25 Falcon flask (passage 2), then progressing to a T75 flask culture was obtained by the third passage."

This advancement holds significance, as it distinguishes itself from the established and widely employed method by Mao et al., where 15 animals are combined to yield a single culture. In contrast, our method achieves a separate culture from each animal, signifying a substantial progression.

3) As claimed (and potentially true) that this method can be applied to isolate TM cells

from other mouse models; authors should demonstrate this using at least with one

other mouse model. This can also demonstrate efficiency of the method as well as

total yield comparison, and Animal reduction aspect too!

Reply: We appreciate your comment. Indeed, it is logical to acknowledge this potential and incorporate it into the manuscript. Our choice to employ albino mice stems (FVB/N) from the fact that in these particular animals, the pigmented circumferential band of TM cannot be observed, and other orientation markers are also more challenging to identify. The outcomes of these experiments have been integrated into the manuscript.

Minor comments:

• TGFB2 exposure and CLAN analysis: Not clear if the % of CLAN positive cells are given as

per 10 coverslips or 10 x 5 = 50 regions observed or per 1274 analyzed cells? What % of

the total cells the 1274 cells represent?

Reply: We suspect that the reviewer has confused two different experiments here. 

1. For each treatment (control or TGFB exposure), five regions per coverslip and 10 coverslips leading to 50 regions were analyzed for the CLAN analysis. 

2. For the percentage of phagocytic cells, 1274 were analyzed as described in Figure 4, of which 83.8% (1068 cells) had taken up fluorobeads within 24h.

• Scale bars need to be defined for figures 4 & 5.

Reply: Thanks very much for this comment. Regrettably, we inadvertently omitted the scale bar in Fig. 4, but we have rectified this error by including it in the figure legend. In Fig. 5, we have denoted that "The scale bar corresponds to 50 µm" only for panel A," The scale bar corresponds to 10 μm" for panel B, and "The scale bar corresponds to 100 µm" for panel D.

We hope that we have adequately answered the reviewers' questions and thank them for their suggestions for improvement, most of which we have incorporated into the manuscript.

Sincerely

Heiko Fuchs

---

## [Editor Report · Decision Letter 1]

7 Dec 2023

A simple dissection method for the isolation of mouse trabecular meshwork cells

PONE-D-23-21644R1

Dear Dr. Fuchs,

We’re pleased to inform you that your manuscript has been judged scientifically suitable for publication and will be formally accepted for publication once it meets all outstanding technical requirements.

Kind regards,

Ashok Kumar, Ph.D.

Academic Editor

PLOS ONE
---

## [Editor Report · Acceptance letter]

11 Dec 2023

PONE-D-23-21644R1 

A simple dissection method for the isolation of mouse trabecular meshwork cells 

Dear Dr. Fuchs:

I'm pleased to inform you that your manuscript has been deemed suitable for publication in PLOS ONE. Congratulations! Your manuscript is now with our production department. 

Kind regards, 

on behalf of

Dr. Ashok Kumar 

Academic Editor

PLOS ONE